# Retrotransposon Protein L1 ORF1p Expression in Aging Central Nervous System

**DOI:** 10.3390/ijms26094368

**Published:** 2025-05-04

**Authors:** Laura Vallés-Saiz, Aaron Abdelkader-Guillén, Jesús Ávila, Félix Hernández

**Affiliations:** Centro de Biología Molecular Severo Ochoa, CSIC/UAM, Universidad Autónoma de Madrid, Cantoblanco, 28049 Madrid, Spain; laura.valles@cbm.csic.es (L.V.-S.); aabdelkader@iib.uam.es (A.A.-G.); javila@cbm.csic.es (J.Á.)

**Keywords:** aging, LINE-1, microglia, ORP1p, retrotransposons

## Abstract

The long-interspersed elements (LINE-1; L1) represent the main active family of retrotransposons in the human organism, comprising approximately 17% of its content. L1 sequence codifies for the two proteins involved in its retrotransposition: ORF1p, an RNA binding protein, and ORF2p, endowed with endonuclease and reverse transcriptase activity. The vast majority of L1 copies are inactive, with only a small percentage retaining their retrotransposition capacity, posing a threat to the organism due to its mutagenic potential. To mitigate such risks, mammals have evolved intricate regulatory mechanisms, including heterochromatin formation and RNA degradation pathways. Age-related diminution in these regulatory pathways may be particularly important within the Central Nervous System (CNS), where cellular regeneration is limited, and genomic integrity is critical for lifelong function. Here, we describe an age-associated upregulation of ORF1p in the mouse brain, indicating a potential role of L1 activity in aging. We further demonstrate the presence of ORF1p across diverse CNS cell types, including neurons, oligodendrocytes and microglia. Notably, we observe a correlation between ORF1p presence and microglial activation, a hallmark of neuroinflammation, during aging. This study advances our understanding of L1 dynamics in the CNS and underscores the significance of L1 in age-related neurological changes.

## 1. Introduction

Transposable elements (TE) consist of sequences of mobile DNA, which have the capacity to propagate throughout the host genome. These elements constitute approximately 45% of the human genome and are generally divided into two primary groups: DNA transposons and RNA transposons, also known as retrotransposons (RTs) [1,2]. DNA transposons mobilize through the cut-and-paste mechanism, but are currently inactive in humans, despite having been essential and highly active during evolution. However, the vast majority of TEs belong to RTs, and propagate via a copy-and-paste mechanism involving RNA intermediates, which are transcribed from the RTs sequence, translated and reverse transcribed into new genomic sites [3,4]. Depending on the presence of flanking repeats, we can distinguish them into long terminal repeats (LTR) and non-LTR retrotransposons (non-LTR). Most of the remaining active RTs are of the latter class, being LINE-1 (long interspersed element-1; L1) family the prime example [4]. Despite nearly 17% of the human genome containing L1 sequences, only a small fraction, around 80–100 copies, maintain their activity due to truncations during retrotransposition, internal rearrangements and mutations [5]. Human retrotransposition-competent L1 is approximately 6 kb in length. Transcription is driven by an RNA polymerase II promoter with a CpG-rich sequence that is usually repressed by DNA methylation. L1 consists of two open reading frames (ORF1 and ORF2) [6], a 5′ untranslated region (UTR) and a 3′ UTR with a poly A tail [7]. ORF1 encodes for an RNA binding protein (ORF1p), while ORF2 encodes for an endonuclease and reverse transcriptase (ORF2p). Both proteins need to interact with the L1 mRNA, forming the L1 ribonucleoprotein (RNP), for retrotransposition to occur [5]. L1 retrotransposition involves several steps. Initially, transcription occurs within the nucleus to produce L1 mRNA, which is then translocated to the cytoplasm for translation. The resulting proteins from this translation are ORF1p and ORF2p, which bind to L1 mRNA to form RNP particles that are transported back to the nucleus. Once in the nucleus, ORF2p, with its endonuclease activity, creates a break in the genomic DNA (gDNA) at a consensus site (5′TTTT/AA3′), and then, with reverse transcriptase activity, uses the exposed poly-T sequence as a primer to generate L1 cDNA, ultimately resulting in the insertion of L1 within the genome.

The ability of RTs to mobilize within the genome represents a significant threat to the host if left unregulated. [8]. Fortunately, organisms have evolved different mechanisms to suppress these potentially harmful elements. Such mechanisms are transcriptional and post-transcriptional silencing [8]. However, the loss of both DNA methylation and repressive histone modifications has been thoroughly reported during aging [9,10,11], which is the foundation of the ‘transposon theory of aging’ [12,13]. Even though RT activation was thought to be restricted to the germline tissues, their expression in somatic tissue has been documented, reshaping our understanding of genomic dynamics [9,14]. The role of RTs has been widely studied in the field of the oncology, its oncogenic potential for tumor progression has been described [7,15]. Despite these advances, the impact of mobile genetic elements such as RTEs on Central Nervous System (CNS) disorders remains a growing area of research. In contrast to previous assumptions that neuronal genomes are static, evidence now shows that L1 elements are not only expressed but are also capable of inserting new copies into the genomes of adult human neurons [16,17].

In the current study, we present findings showing an association between an upregulation of ORF1p, derived from L1, and the aging process in the murine CNS. We document an increase in L1 transcription correlating with age, and our findings contribute to the growing body of evidence that suggests a significant role of L1 in the aging process.

## 2. Results

### 2.1. Characterization of the Presence of ORF1p in the Brain

Earlier studies indicate that although the expression of transposable elements (TEs) is generally low in somatic tissues, it tends to increase with age [18]. In light of this, we assessed the presence of L1 in the mouse brain by immunofluorescence with an antibody that recognizes ORF1p from L1. Initially, we tested different brain regions for the presence of ORF1p as: CA1 hippocampal region, dentate gyrus, somatosensory cortex and entorhinal cortex (Figure 1). The highest expression of ORF1p was observed in the entorhinal cortex, so this region was chosen as our main region of study for all subsequent experiments.

Focusing on the entorhinal cortex, we identified different staining patterns, which we classified into two distinct types of ORF1p staining. We referred to them as Type I, characterized by intense labeling as present mainly in old mice, and Type II, characterized by diffuse labeling and present mainly in young mice (Figure 2). Afterwards, we conducted a quantification analysis of cells exhibiting ORF1p staining, aiming to investigate the aging process across age groups of 3, 6, 12, 18 and 24 months. We found significant differences between young and aged groups for Type I staining (Figure 2B) (i.e., between 3 and 24 months old (* *p* < 0.001) and between 6 and 24 months old (* *p* < 0.001), among others), but not for type II staining (Figure 2C). Based on these results, it appears that ORF1p Type I labeling serves as a promising biomarker for aging.

### 2.2. ORF1p Distribution Through Brain Cell Types

As we observed in our previous analysis, ORF1p expression seems to be enriched in specific brain regions, such as the entorhinal cortex, which is critical for memory formation and navigation [19]. To further explore the cellular specificity of ORF1p expression, we extended our investigation to various types of neuroglial cells in the brain. Our aim was to determine not just where but also in which type of cells ORF1p is expressed. This would provide insight into the potential functional implications of ORF1p in different cellular contexts within the CNS.

We conducted a detailed analysis involving the co-staining of ORF1p with several well-established neural and glial cell markers. Specifically, we examined its presence in conjunction with NeuN, a widely recognized marker for mature neurons [20], Iba1, a marker for microglia [21] and Olig2, a marker for oligodendrocytes [22] (Figure 3). Interestingly, while we observed colocalization of ORF1p with markers for neurons, microglia and oligodendrocytes, indicating its expression across these diverse cell types, we did not observe any colocalization with GFAP, a marker for astrocytes (Figure 3).

### 2.3. ORF1p in Microglial Activation During Aging

As previously shown, ORF1p expression is taking place in microglia. Previous studies reported that microglial activation is increased during aging [23,24]. Additionally, the activation of another type of TEs, specifically endogenous retroviruses has been associated with the induction of an inflammatory response [25]. Given these findings, we hypothesized that L1 might also be related to this phenomenon. To investigate this possibility, we employed CD68, a lysosomal protein expressed in high levels by activated microglia and in low levels by resting microglia [26], along with Iba1 as a total microglial marker [21]. We performed a quantification of triple positive cells for CD68/Iba1/ORF1p Type I (Figure 4A) and established a ratio by dividing this number by the total Iba1^+^ cells. We do not observe a correlation between the 3 months old and aged groups. Moreover, significant differences were observed between 3 and 6 months old groups (* *p* < 0.05). However, a significant colocalization of ORF1p in activated microglia between young and aged groups, specifically between 6 and 24 months old (**** *p* < 0.0001) and between 6 and 18 months old mice (** *p* < 0.01) was found (Figure 4B). These findings suggest a potential role of L1 in microglial aging activation.

## 3. Discussion

The “heterochromatin loss” model of aging, proposed by Villeponteau in 1997 [27], suggests that during the aging process there is a reduction in heterochromatin levels. This reduction can lead to the activation of previously inactive or untranscribed genetic regions. Therefore, regions that were not previously involved in gene expression may begin to be transcribed, potentially affecting cellular processes and the aging phenotype [27]. An example of this phenomenon is the expression of transposable elements (TEs) (for a review see [28]). It is important to note that, as individuals age, the regulatory mechanisms that suppress these elements weaken, which could disrupt homeostasis and increase the incidence of diseases [10,11]. This decline serves as the foundation of the ‘transposon theory of aging’, which proposes that epigenetically silenced transposable elements become reactivated as regulatory controls diminish with age [12,13]. In order to confirm this idea in the CNS, we decided to examine L1 retrotransposon, the most abundant and active transposable element in humans. While nearly 17% of the human genome derives from L1 sequences, almost all of them are remnants of their evolutionary role. Nonetheless, around 80–100 L1 copies remain capable of retrotransposition, maintaining the ability to express ORF1p and ORF2p and mobilize within the genome [7,29]. L1 transposition was observed within CNS [16,17], which gives rise to the idea that transposable elements, especially retrotransposons, could be more implicated in CNS homeostasis than previously appreciated. In fact, L1 promoters are dynamically active in the developing and the adult human brain [30].

Here, we sought to explore whether the ORF1p could be a biomarker of aging in the CNS by using wild-type mice of different ages. Previous studies suggest a RT presence in the CNS and their possible implications in aging [16,29]. These studies explored the presence of retrotransposon L1 across various biological contexts, none have directly examined L1 ORF1p via immunofluorescence within the aged mouse brain. In our study, we characterized ORF1p immunostaining in the CNS, classifying it into two groups: Type I and Type II. Type I labeling appears as high levels of ORF1p or likely aggregated. It could be that only phosphorylated and aggregated ORF1p serves as a biomarker of aging, as phosphorylation followed by aggregation is essential for ORF1p activity and L1 retrotransposition [31,32]. Interestingly, these aggregates are similar to large cytoplasmic compartments enriched in ORF1p, L1 mRNA and ribosomes found in BALB/c wild-type spermatocytes and termed as L1 bodies [33].

Conversely, Type II exhibits a diffuse, speckled staining, typically seen in young mice. We noted that the entorhinal cortex is the brain region with the highest number of ORF1p-positive cells for both staining patterns. Our findings indicate a significant association between Type I ORF1p staining and the aging process in the entorhinal cortex. This observation could be relevant as the entorhinal cortex is the first region showing histological alterations in Alzheimer’s disease (AD) [34].

Increased ORF1p raises questions about its potential impact on neuronal function, which could be pivotal in understanding age-related changes. However, the underlying mechanisms of this association remain unclear. It is uncertain whether these changes are attributable to increased transcription and translation activities or a deficiency in ORF1p degradation pathways. Also, considering the presence of ORF1p in the CNS, one possibility is that such machinery encoded by L1 transposons may also assist SINE transposons that are entirely dependent on the machinery encoded by LINE-1 to initiate their transposition processes [35,36].

The expression of L1-encoded proteins seems to be independent of L1 RNA levels. Thus, in a Huntington’s disease (HD) mouse model, differential expression of ORF1p and ORF2p was observed in specific brain regions and at certain ages, suggesting activation of the retrotransposition machinery [37]. Remarkably, older HD mice (24 months) exhibited increased ORF1p levels coinciding with neurodegeneration and reactive gliosis.

In addition to its role in aging, the L1 retrotransposon and its encoded ORF1p protein can contribute to neurodegeneration through multiple interconnected mechanisms. Substantial evidence indicates that L1 retrotransposition induces genomic instability [38], promotes age-associated inflammation through interferon-mediated responses [39], and L1 activation promotes oxidative stress that exacerbates neuronal damage [12].

Following this first characterization, we explored ORF1p expression throughout different brain cell types. Our findings confirm ORF1p expression in neurons, oligodendrocytes and microglia, but not in astrocytes. This absence of ORF1p expression in astrocytes could be due to a cell-type specific L1 expression. The lack of ORF1p expression in astrocytes implies a selective function or mechanism in other cell types. This observation could hold considerable significance in comprehending the molecular underpinnings of diseases, particularly those where these cell types are affected differently.

Microglia plays a very important role in maintaining brain homeostasis. As aging progresses, there is a notable increase in the activation of microglia, leading to heightened inflammatory signaling in the aged brain [23]. Since we observed ORF1p expression in microglia, we considered it of great interest to examine the existence of a correlation between microglia priming through aging and ORF1p expression. Our results indeed show that correlation, suggesting L1 may be involved in this priming process, either as a cause or as a consequence. Interestingly, elevated expression of the retrotransposon LINE-1 drives Alzheimer’s disease-associated microglial dysfunction [40]. Additionally, we observed a great ratio of active microglia in the youngest mice, which was unexpected. At the moment, our only explanation would be an activation of microglia due to the process of synaptic pruning. However, in mice, synaptic pruning occurs in the first few weeks of life [41]. Future experiments in the field should examine the mechanisms underlying these observations and whether aging-associated de-repression of RTs might play a more central role in the matter than previously thought.

In summary, current evidence supports the relevant role of retrotransposons (especially L-1), and particularly ORF1p, in brain aging processes and, likely, in the pathophysiology of neurodegenerative diseases. Thus, it has been recently described that LINE-1 ORF1p expression is higher in microglia from “late onset Alzheimer’s disease (LOAD)” patients compared to controls [40]. The reactivation of L1 in adult neurons and its elevated expression in vulnerable regions such as the entorhinal cortex, as well as in microglia, suggest direct involvement in the neuro-inflammation and cellular damage associated with these pathologies. Furthermore, pharmacological inhibition of ORF2p has shown promising effects by reducing markers of neurodegeneration and inflammation in preclinical models [42], opening up new therapeutic prospects. Although this strategy has not yet been shown to completely reverse neurodegenerative processes, it does appear to slow their course and offer neuronal protection. These approaches still require further evaluation of their long-term efficacy and safety, although they position transposable elements as emerging targets with potential in the treatment of central nervous system diseases.

## 4. Materials and Methods

### 4.1. Animals and Tissue Processing

All mice used were housed at the “Centro de Biología Molecular Severo Ochoa” animal facility, with access to food and water ad libitum in a colony kept at 19–22 °C and 40–60% humidity, under a 12–12 h light/dark cycle. Three different mice strains were used: 3-, 6-, 12-, 18- and 24-month-old male and female wild-type mice (C57BL/6J). The study was carried out in accordance with the ARRIVE guidelines and all methods were performed in accordance with the relevant guidelines and regulations. It was approved by the CBMSO’s (AECC-CBMSO-13/172) and national (PROEX 102.0/21) Ethics Committees.

Animal strains were deeply anesthetized and perfused transcardially with 0.9% saline solution prior to brain retrieval. One brain hemisphere was dissected into main brain regions, snap-frozen in liquid nitrogen and stored at −80 °C. The other hemisphere was fixated in 4% paraformaldehyde (PFA) in 0.1 N phosphate buffer (PB) for 24 h at 4 °C. Subsequently, the samples were washed three times in PB and placed in a 10% sucrose/4% agarose matrix. Sagittal sections were obtained on a Leica VT1200S vibratome (50 μm) and preserved in cryoprotective solution.

### 4.2. Immunofluorescence

Brain sections were washed 3 times with PB 0.1 N and incubated for 24 h at 4 °C with the following primary antibodies in blocking solution (1% BSA and 1% Triton X-100 in PB 0.1 N): anti-GFAP (chicken, 1:2000, Abcam AB4674, Cambridge, UK), anti-Iba1 (guinea pig, 1:500, Synaptic Systems #234004, Göttingen, Germany), anti-Olig2 (goat, 1:1000, RD Systems AF2418, Minneapolis, MN, USA), anti-NeuN (mouse, 1:1000, Millipore AB377, Burlington, MA, USA), anti-ORF1 (rabbit, 1:250 and 1:400, Abcam AB216324), anti-CD68 (rat, 1:500, Abcam AB53444). Afterward, the sections underwent 5 washes using the identical blocking solution and were incubated for 24 h at 4 °C with the corresponding secondary antibodies conjugated with Alexa fluorophores in blocking solution. Three additional washes were performed with PB 0.1 N, followed by a 10 min incubation with 1 μg/mL DAPI (Sigma-Aldrich D9542, St. Louis, MO, USA). Finally, sections were washed three times with PB 0.1 N and placed on gelatinized slides utilizing FluorSave reagent.

### 4.3. Image Acquisition and Analysis

Images were captured using an Olympus SpinSR10 spinning disk confocal microscope with a 60X objective lens. Each image represents an overlay of 101 stacks. each 0.2 μm thick. Background noise was eliminated prior to manual cell counting using Fiji Cell Counter tool. Additionally, an invariable threshold was set using the Triangle algorithm for intensity and area measurements. The representative images in this manuscript comprise a Z-stack composed of 15 of these stacks.

### 4.4. Statistical Analysis

Data are presented as mean ± SEM. Comparisons of means across three or more groups were conducted using ordinary one-way ANOVA, except where noted otherwise. For comparisons between two groups, two-tailed unpaired *t*-tests were utilized. All statistical analyses were carried out using Prism 8 software by GraphPad.

## Figures and Tables

**Figure 1 ijms-26-04368-f001:**
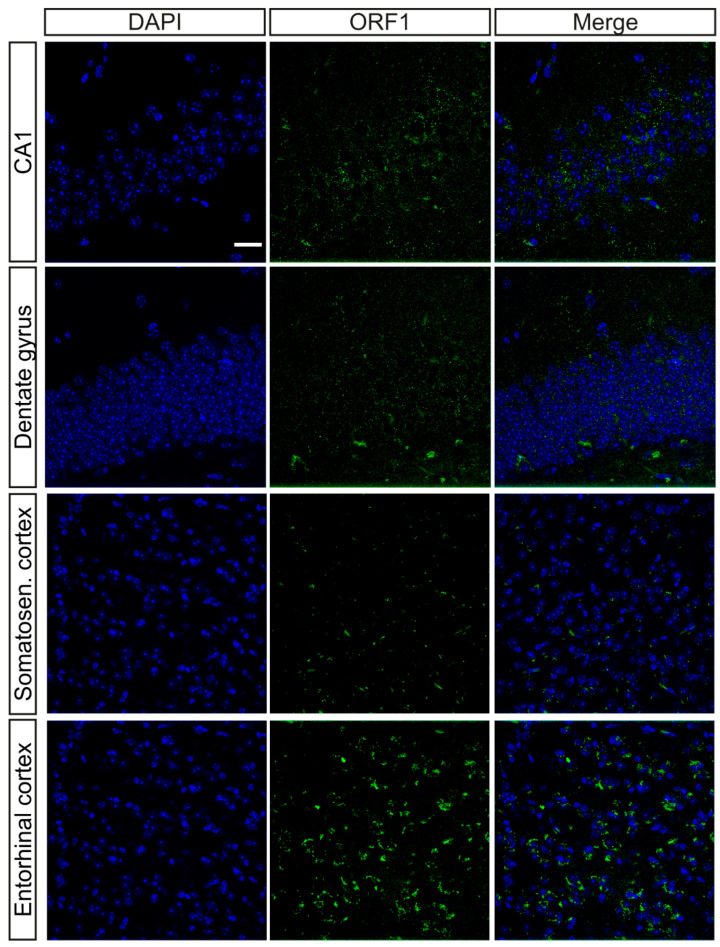
Characterization of ORF1p presence in brain. Representative immunofluorescence images of ORF1p labeling (green) in CA1, dentate gyrus, somatosensory cortex and entorhinal cortex in 18-month-old wild-type mice. DAPI was used as nucleus marker (blue). Scale bar: 20 μm.

**Figure 2 ijms-26-04368-f002:**
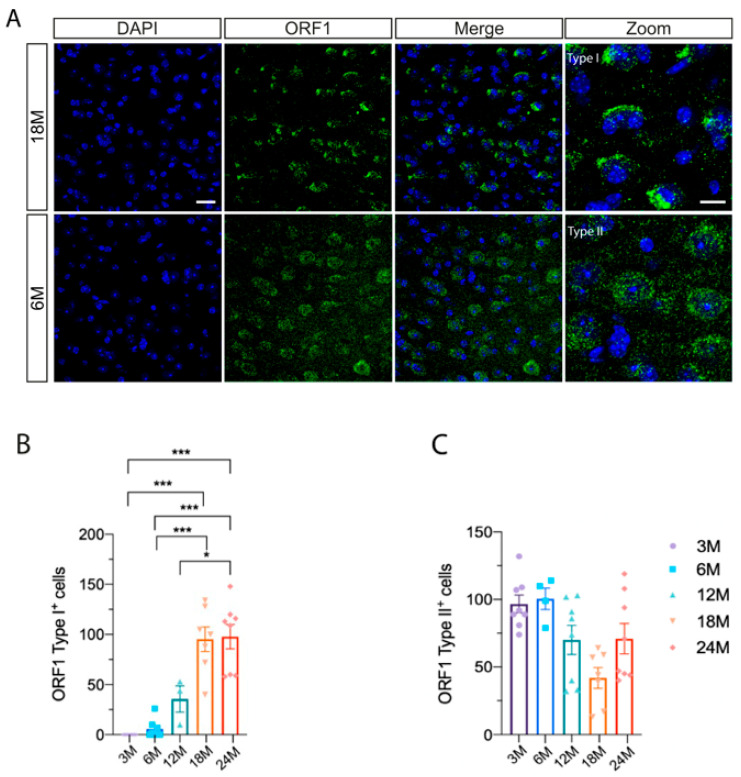
Quantification of ORF1p presence in brain. (**A**) Representative images for entorhinal cortex in 18- and 6-mounth old mice. Type I (intense labeling) and II (diffuse labeling) ORF1p staining (green) is observed in old and young mice, respectively. Scales bar: 20 and 10 μm (zoom). (**B**) Quantification of ORF1p Type I^+^ and (**C**) quantification of ORF1p Type II^+^ cells per section in entorhinal cortex during aging (n = 4–5 per age). * *p* < 0.05, *** *p* < 0.001.

**Figure 3 ijms-26-04368-f003:**
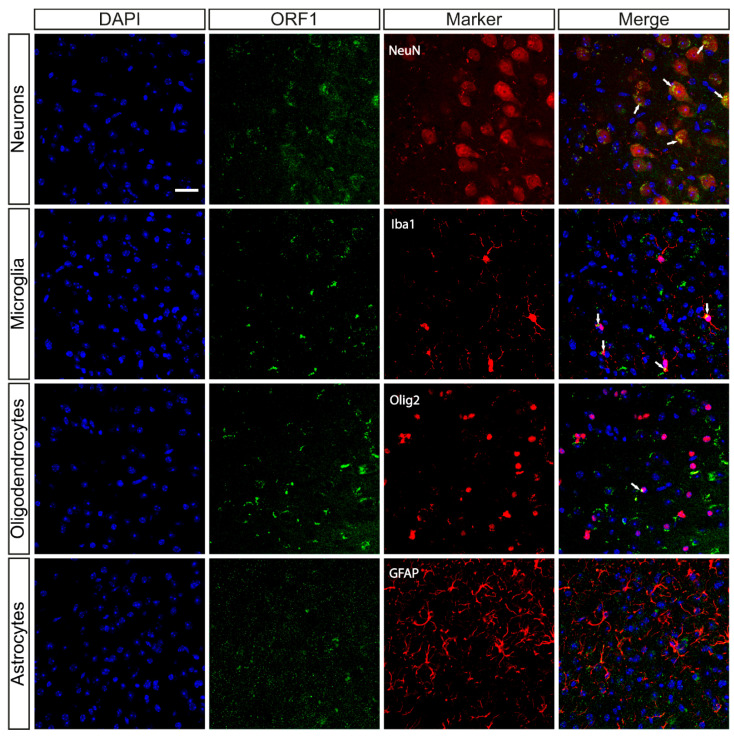
Colocalization of ORF1p and specific brain cell type markers. Representative immunofluorescence images of the ORF1p marker (green) and various neuroglial markers such as: NeuN (neurons), Iba1 (microglia), Olig2 (oligodendrocytes) and GFAP (astrocytes) (all of them in red) in 12-month-old wild type mice. The merge between ORF1p and neuroglial marker is shown to observe possible colocalization. Colocalization is marked by white arrows. DAPI was used as nucleus marker (blue). Scale bar: 20 μm.

**Figure 4 ijms-26-04368-f004:**
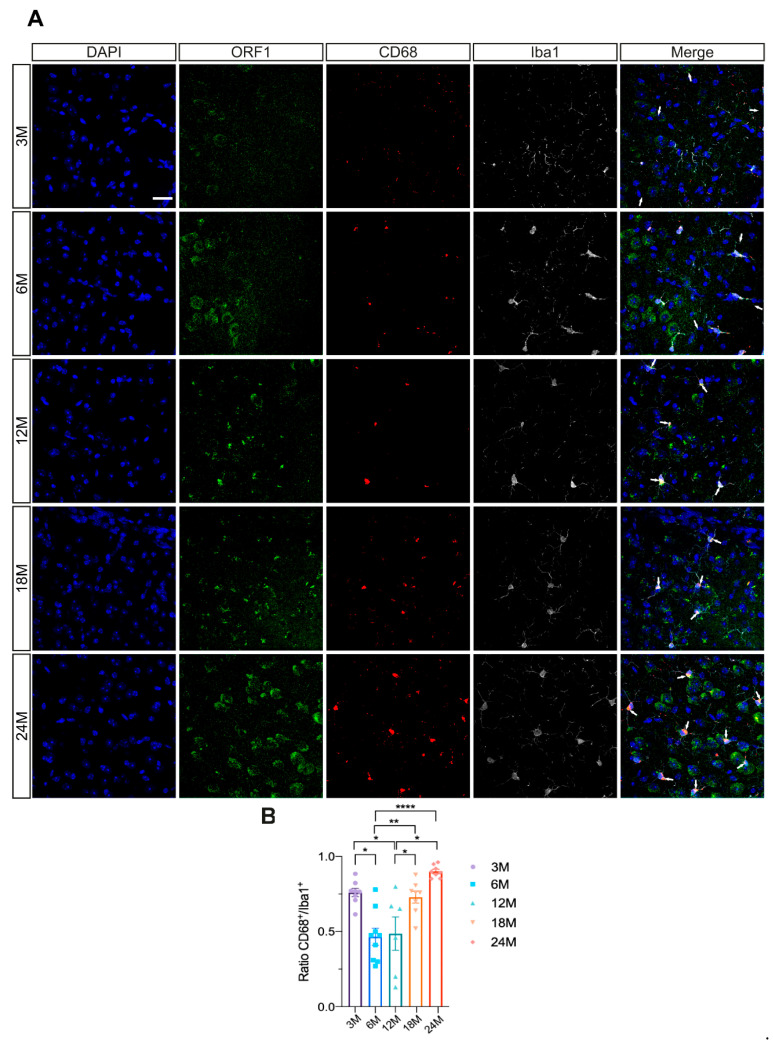
Microglial activation and ORF1p. (**A**) Representative immunofluorescence images of the ORF1p marker (green), activated microglia CD68 (red) and total microglial marker Iba1 (gray) in entorhinal cortex during aging in wild-type mice. The merge between ORF1p and microglial markers is shown to observe possible colocalization. Scale bar: 20 μm (**B**) Ratio of Iba1 cells per section showing colocalization of ORF1p type I and CD68. Error bars represent the standard error of the mean, and significant differences were determined by unpaired ANOVA one-way test (n = 5–6 per age). * *p* < 0.05, ** *p* < 0.01, **** *p* < 0.0001.

## Data Availability

The datasets generated during and/or analyzed during the current study are available from the corresponding author on reasonable request.

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
