# Peer review of "Retrotransposon Protein L1 ORF1p Expression in Aging Central Nervous System"

_ijms, 2025, doi:10.3390/ijms26094368_

Round 1

Reviewer 1 Report

Comments and Suggestions for Authors

This manuscript addresses the expression of ORF1p, encoded by L1 RNAs, in the mouse brain during aging.
They found an age-associated upregulation of ORF1p and they proved its expression in neurons, oligodendrocytes and microglia. They also observed a correlated pattern of ORF1p expression and microglial activation, a hallmark of neuroinflammation, during aging. 
They hypothesize that this increased expression is involved in aging.

Major points:

  1. Immunochemistry of L1-derived proteins is tricky. Authors use a single Ab commercially produced by Santacruz. If not present in the literature, a characterization of the specificity of the antibody should be shown. A second Ab for ORF1p should be used to validate their data.

  2. Double labeling should be carried out to distinguish Type I from Type II staining.

  3. Given the high number of cells that showed a potential increase of ORF1p, a time course analyzed with western blot should be sensitive enough to provide further evidence.

In addition:
The discussion should be improved. There is no mention of various reports proving that there is no full correspondance between ORFp1 and ORF2p expression. If there is no ORF2p what could ORF1p function be?
Proving increased expression of ORF1p in aging is not new (see i.e. Floreani et al Front. Cell. Neurosci 2022). A detailed analysis of data from the literature should be discussed.

Author Response

Reviewer 1:

This manuscript addresses the expression of ORF1p, encoded by L1 RNAs, in the mouse brain during aging.

They found an age-associated upregulation of ORF1p and they proved its expression in neurons, oligodendrocytes and microglia. They also observed a correlated pattern of ORF1p expression and microglial activation, a hallmark of neuroinflammation, during aging.

They hypothesize that this increased expression is involved in aging.

Major points:

Immunochemistry of L1-derived proteins is tricky. Authors use a single Ab commercially produced by Santacruz. If not present in the literature, a characterization of the specificity of the antibody should be shown. A second Ab for ORF1p should be used to validate their data.

Anti-LINE-1 ORF1p antibody (Abcam AB216324] is a rabbit recombinant monoclonal antibody designed to detect the LINE-1 open reading frame 1 protein (ORF1p). Validated for immunocytochemistry/immunofluorescence, western blot (WB), immunohistochemistry (IHC), and flow cytometry and suitable for mouse samples. This antibody has been used at least in 12 publications some of them focused in Central Nervous System (PubMed 36207411; PubMed 36070749). In addition, in the Materials and Methods section of the manuscript we have perfectly identified the antibody so that researchers know its reference (Line 275).

Double labeling should be carried out to distinguish Type I from Type II staining.

We understand the importance of distinguishing type I from type II cells. However, immunohistochemistry (IHC) with the anti-LINE-1 ORF1p antibody cannot differentiate two signals obtained with the same antibody when a second antibody is unavailable. This is possible, for example, in Alzheimer's disease, where antibodies exist that discriminate between unphosphorylated, phosphorylated, or aggregated tau protein. We are currently analyzing the ORF1p signal obtained in cells with type I staining to determine whether they represent aggregated forms or are capable of sequestering other proteins or nucleic acids. However, we believe that such analysis is beyond the scope of this work.

Given the high number of cells that showed a potential increase of ORF1p, a time course analyzed with western blot should be sensitive enough to provide further evidence.

We have performed Western-blots but the signal was too weak and not conclusive results were obtained. We do believe that immunohistochemistry results demonstrate its expression in neurons, oligodendrocytes and microglia.

In addition:

The discussion should be improved. There is no mention of various reports proving that there is no full correspondance between ORFp1 and ORF2p expression. If there is no ORF2p what could ORF1p function be? Proving increased expression of ORF1p in aging is not new (see i.e. Floreani et al Front. Cell. Neurosci 2022). A detailed analysis of data from the literature should be discussed.

1) We have added this reference and commented the concerns raised by the reviewer as follow (Lines 207-216):

“The expression of L1-encoded proteins seems to be independent of L1 RNA levels. Thus, in a Huntington’s disease (HD) mouse model, differential expression of ORF1p and ORF2p was observed in specific brain regions and at certain ages, suggesting activation of the retrotransposition machinery [37]. Remarkably, older HD mice (24 months) exhibited increased ORF1p levels coinciding with neurodegeneration and reactive gliosis.

In addition to its role in aging, the L1 retrotransposon and its encoded ORF1p protein can contribute to neurodegeneration through multiple interconnected mechanisms. Substantial evidence indicates that L1 retrotransposition induces genomic instability [38], promotes age-associated inflammation through interferon-mediated responses [39], and L1 activation promotes oxidative stress that exacerbates neuronal damage [12].”

2) A new sentence and reference has been added (Line 231-232). “Interestingly, elevated expression of the retrotransposon LINE-1 drives Alzheimer’s disease-associated microglial dysfunction [40].”

Reviewer 2 Report

Comments and Suggestions for Authors

Reviewer:

This study identifies an age-related increase in ORF1p, a key L1 protein, in the mouse brain. ORF1p is detected in various CNS cell types, including neurons, oligodendrocytes, and microglia. Additionally, its presence correlates with microglial activation, a marker of neuroinflammation in aging. These findings highlight L1's potential role in age-related neurological changes. While this study is intriguing and well-written, several important questions require the author's attention.

  1. The author needs to identify the role or function of ORF1p in the aging brain. What is the role of ORF1p in these cells? Whether Knockout or overexpression ORF1p can delay or accelerate aging?
  2. How do I know this antibody of ORF1p is specific to recognize ORF1p? The author should perform immunostaining on ORF1p knockdown or knockout cells or tissues to confirm the specificity of the ORF1p antibody. 

Author Response

Reviewer 2:

This study identifies an age-related increase in ORF1p, a key L1 protein, in the mouse brain. ORF1p is detected in various CNS cell types, including neurons, oligodendrocytes, and microglia. Additionally, its presence correlates with microglial activation, a marker of neuroinflammation in aging. These findings highlight L1's potential role in age-related neurological changes. While this study is intriguing and well-written, several important questions require the author's attention.

The author needs to identify the role or function of ORF1p in the aging brain. What is the role of ORF1p in these cells? Whether Knockout or overexpression ORF1p can delay or accelerate aging?

To answer the question about the function of ORF1p in the aging brain, we have rewritten and included the following paragraphs:

1) (Lines 207-216):

“The expression of L1-encoded proteins seems to be independent of L1 RNA levels. Thus, in a Huntington’s disease (HD) mouse model, differential expression of ORF1p and ORF2p was observed in specific brain regions and at certain ages, suggesting activation of the retrotransposition machinery [37]. Remarkably, older HD mice (24 months) exhibited increased ORF1p levels coinciding with neurodegeneration and reactive gliosis.

In addition to its role in aging, the L1 retrotransposon and its encoded ORF1p protein can contribute to neurodegeneration through multiple interconnected mechanisms. Substantial evidence indicates that L1 retrotransposition induces genomic instability [38], promotes age-associated inflammation through interferon-mediated responses [39], and L1 activation promotes oxidative stress that exacerbates neuronal damage [12].”

2) (Lines 238-252).

“In summary, current evidence supports the relevant role of retrotransposons (especially L-1), and particularly ORF1p, in brain aging processes and, likely, in the pathophysiology of neurodegenerative diseases. Thus, it has been recently described that LINE-1 ORF1p expression is higher in microglia from LOAD patients compared to controls [37]. The reactivation of L1 in adult neurons and its elevated expression in vulnerable regions such as the entorhinal cortex, as well as in microglia, suggest a direct involvement in the neuroinflammation and cellular damage associated with these pathologies. Furthermore, pharmacological inhibition of ORF2p has shown promising effects by reducing markers of neurodegeneration and inflammation in preclinical models [38], opening up new therapeutic prospects. Although this strategy has not yet been shown to completely reverse neurodegenerative processes, it does appear to slow their course and offer neuronal protection. These approaches still require further evaluation of their long-term efficacy and safety, although they position transposable elements as emerging targets with potential in the treatment of central nervous system diseases.”

How do I know this antibody of ORF1p is specific to recognize ORF1p? The author should perform immunostaining on ORF1p knockdown or knockout cells or tissues to confirm the specificity of the ORF1p antibody.

About ORF1p knockdown or knockout cells, LINE1 RNA knockdown strategy using antisense oligos (ASOs) has been carried out in in mouse embryonic stem cells (https://doi.org/10.1016/j.cell.2018.05.043). However, we believe this is beyond the scope of the work presented here. Such an approach can be performed in cultured cells, but it is difficult in an animal model as complex as the mouse.

In addition, the Anti-LINE-1 ORF1p antibody (Abcam AB216324) is a recombinant rabbit monoclonal antibody developed to recognize the LINE-1 open reading frame 1 protein (ORF1p). It has been validated for use in immunocytochemistry/immunofluorescence, western blot (WB), immunohistochemistry (IHC), and flow cytometry, and is compatible with mouse-derived samples. This antibody has been cited in at least 12 scientific publications, including studies focused on the central nervous system (e.g., PubMed IDs: 36207411 and 36070749). Furthermore, the antibody is clearly identified in the Materials and Methods section of the manuscript (Line 275), providing researchers with its exact reference.

Round 2

Reviewer 2 Report

Comments and Suggestions for Authors

The author addressed all my concerns. 

Author Response

We thank the reviewer for his comments and are pleased to know that our responses to them have been satisfactorily answered.